# Transcriptional and Translational Dynamics of Zika and Dengue Virus Infection

**DOI:** 10.3390/v14071418

**Published:** 2022-06-28

**Authors:** Kamini Singh, Maria Guadalupe Martinez, Jianan Lin, James Gregory, Trang Uyen Nguyen, Rawan Abdelaal, Kristy Kang, Kristen Brennand, Arnold Grünweller, Zhengqing Ouyang, Hemali Phatnani, Margaret Kielian, Hans-Guido Wendel

**Affiliations:** 1Cancer Biology and Genetics Program, Memorial Sloan-Kettering Cancer Center, New York, NY 10065, USA; wendelh@mskcc.org; 2Department of Molecular Pharmacology, Albert Einstein College of Medicine, Albert Einstein Cancer, Center, Bronx, NY 10461, USA; tranguyen.nguyen@einsteinmed.edu; 3Department of Cell Biology, Albert Einstein College of Medicine, Bronx, NY 10461, USA; mguamartinez@gmail.com (M.G.M.); rawanjamil@gmail.com (R.A.); margaret.kielian@einsteinmed.org (M.K.); 4Global Innovation, Boehringer Ingelheim Animal Health, 69800 Saint-Priest, France; 5The Jackson Laboratory for Genomic Medicine, Farmington, CT 06032 and Department of Biomedical Engineering, University of Connecticut, Storrs, CT 06269, USA; jianan.jay.lin@gmail.com; 6Department of Neurology, Vagelos College of Physicians & Surgeons of Columbia University, New York, NY 10032, USA; james.alan.gregory@gmail.com (J.G.); kkang@nygenome.org (K.K.); hphatnani@nygenome.org (H.P.); 7Center for Genomics of Neurodegenerative Disease, New York Genome Center, New York, NY 10013, USA; 8Division of Molecular Psychiatry, Departments of Psychiatry and Genetics, Yale School of Medicine, New Haven, CT 06510, USA; kristen.brennand@gmail.com; 9Institute of Pharmaceutical Chemistry, Philipps University Marburg, 35032 Marburg, Germany; arnold.gruenweller@staff.uni-marburg.de; 10Department of Biostatistics and Epidemiology, School of Public Health and Health Sciences, University of Massachusetts, Amherst, MA 01003, USA; ouyang@schoolph.umass.edu

**Keywords:** ZIKV, polyamine pathways, ATF3/CHOP, BACH1/2, dengue, eIF5A hypusination, PIM1

## Abstract

Zika virus (ZIKV) and dengue virus (DENV) are members of the Flaviviridae family of RNA viruses and cause severe disease in humans. ZIKV and DENV share over 90% of their genome sequences, however, the clinical features of Zika and dengue infections are very different reflecting tropism and cellular effects. Here, we used simultaneous RNA sequencing and ribosome footprinting to define the transcriptional and translational dynamics of ZIKV and DENV infection in human neuronal progenitor cells (hNPCs). The gene expression data showed induction of aminoacyl tRNA synthetases (ARS) and the translation activating PIM1 kinase, indicating an increase in RNA translation capacity. The data also reveal activation of different cell stress responses, with ZIKV triggering a BACH1/2 redox program, and DENV activating the ATF/CHOP endoplasmic reticulum (ER) stress program. The RNA translation data highlight activation of polyamine metabolism through changes in key enzymes and their regulators. This pathway is needed for eIF5A hypusination and has been implicated in viral translation and replication. Concerning the viral RNA genomes, ribosome occupancy readily identified highly translated open reading frames and a novel upstream ORF (uORF) in the DENV genome. Together, our data highlight both the cellular stress response and the activation of RNA translation and polyamine metabolism during DENV and ZIKV infection.

## 1. Importance

Zika and dengue virus are major causes of morbidity in tropical countries and with a changing climate, they are increasingly seen across the globe. Zika and dengue virus cause common and unique pathological symptoms. It is important to understand the molecular biology of these viruses and pinpoint the common and specific effects of the two viruses on the gene expression programs in the host genome. In this study, we present a comprehensive data resource encompassing gene expression and mRNA translation during ZIKV and DENV infection. Transcription data indicate distinct cell stress responses triggered by DENV and ZIKV infection. We observe translation activation by PIM1 kinase and induction of aminoacyl tRNA synthetases (ARS). Translation of cellular mRNAs encoding key polyamine and mRNA translation factors is increased upon virus infection. We further define open reading frames of ZIKV and DENV RNA based on ribosome occupancy.

## 2. Introduction

Zika virus (ZIKV) and dengue virus (DENV) belongs to the family *Flaviviridae* and contain single-stranded plus-sense RNA genomes [1]. Both viruses are transmitted to humans by tropical *Aedes* mosquitos that are increasingly reaching first world regions. There have been frequent ZIKV outbreaks since 2007 and infections are linked to multi-organ failure in adults, and fetal defects including microcephaly, other malformations, and fetal demise [2,3]. While still rare in the USA, DENV is one of the worst mosquito-borne human pathogens in the world. DENV infections have increased dramatically and currently stand at 100–400 million infections per year. This makes DENV a leading cause of disease in tropical countries [4]. Clinical symptoms of DENV infection have led to the name “breakbone fever” and severe cases include hemorrhagic fever, dengue shock syndrome, and death [5]. A live-attenuated DENV vaccine has been approved in 2015 and complements mosquito control measures and personal protection (World Health Organization) [6].

ZIKV and DENV share over 90% of their genome sequences but show differences in cellular tropism and molecular effects. Previous studies on the transcriptional and immunological effects of DENV and ZIKV have revealed that both viruses induce a classical Type I interferon anti-viral response [7]. Single-cell sequencing revealed activation of an MX2-related transcription program in the B lymphocytes [8]. Transcriptome meta-analysis in DENV infected human patients and human monocyte THP-1 cells revealed effects on cell junction and extracellular matrix proteins [9]. The transcriptional effects of ZIKV infection appear distinct and changes in cell cycle, mRNA processing, and metabolism have been reported [10,11]. Both viruses also modulate mRNA translation and their replication and translation is a potential vulnerability [12]. For example, ribosome profiling in liver cancer cells suggests that DENV activates ER-linked cellular stress while the effects of ZIKV are not known [13]. To enhance their replication and translation, both viruses mimic the 5′UTR cap structure of cellular mRNAs and escape the cellular recognition [14,15,16]. However, they depend on cellular enzymes such as the eIF4E kinase p38-MNK1 and the RNA helicase eIF4A (DDX2) and both enzymes are accessible with current inhibitors [17,18,19,20]. In this study, we use high-resolution ribosome profiling and RNA deep sequencing (RNA-seq) to define the gene expression and mRNA translation dynamics of the viral and host genomes during ZIKV and DENV infection of human neuronal progenitor cells (hNPCs).

## 3. Materials and Methods

### 3.1. Cell Culture and Virus Infection

We used primary hNPCs obtained from two different human donors (clone A and clone B). All hiPSC research was conducted under the oversight of the Institutional Review Board (IRB) and Embryonic Stem Cell Research Overview (ESCRO) committees at ISSMS. The participants provided written informed consent. NPCs were seeded onto matrigel-coated tissue culture plates at 300k cells per well of a 6-well plate in NPC medium (DMEM/F12 (Life Technologies, Carlsbad, CA, USA, #10565) supplemented with 1 × N2 (Life Technologies, #17502-048), 1 × B27-RA (Life Technologies, #12587-010), 20 ng/mL FGF2 (R&D, #233-FB-10), and 1 mg/mL Natural Mouse Laminin (Life Technologies, #23017-015). NPCs were fed every second day and split once per week using Accutase. ZIKV IbH 30656 NR-50066 was isolated from the blood of a human in Ibadan, Nigeria, and was obtained by M.K. through BEI Resources, NIAID, and NIH, as part of the WRCEVA program. DENV-2 strain 16,681 was obtained by M.K. from the World Arbovirus Reference Center, University of Texas Medical Branch, Galveston, TX. A total of 6-well plates were coated with poly-L-Ornithine (PLO) and laminin before seeding hNPCs (8 × 10^5^ cells per well) [21]. During 24 h post-plating, the cells were infected with ZIKV or DENV at a multiplicity of infection of 1 and incubated at 37 °C. Viral inocula were removed at 3 h post-infection (hpi). Cells were collected and processed for ribosome footprinting at 72 hpi.

### 3.2. Ribosome Footprinting

Human neuronal progenitor cells (hNPCs) (n = 2) were infected with ZIKV-Ibh and DV-2 16,681 (72 h) followed by cycloheximide treatment for 10 min. Total RNA and ribosome-protected fragments were isolated following the published protocol [22]. Small RNA libraries were generated using the SMARTer smRNA kit from Illumina. Deep sequencing libraries were generated from two independent clones in replicates (n = 2) and sequenced on the HiSeq 2000 platform. Genome annotation was from the human genome sequence GRCh37 downloaded from Ensembl public database: http://www.ensembl.org, accessed on 17 October 2018. For the virus genome, we used the reference genomes for ZIKV-Ibh and DV-2 16,681 downloaded from the NCBI database [23].

### 3.3. Sequence Alignment

First, ribosome footprint (RF) reads were filtered based on the quality score, which kept reads that have a minimum quality score of 25 for at least 75 percent of the nucleotides. Second, the linker sequence (5′- CTGTAGGCACCATCAAT-3′) was trimmed from the 3′ end of the reads. Next, we filtered out the reads shorter than 15nt after the linker-trimming step. All these steps were done by using FASTX-Toolkit (http://hannonlab.cshl.edu/fastx_toolkit/index.html accessed on 17 October 2018). The ribosome footprint reads were first aligned to the virus genome using Bowtie2 [24]. Specifically, the reads were first mapped to ZIKV and DENV genomes. The unmapped reads were used for downstream analysis of the human genome. To remove ribosomal RNA, the footprint reads were then aligned to the ribosome RNA sequences of GRCh37 downloaded from UCSC Table Browser (https://genome.ucsc.edu/cgi-bin/hgTables accessed on 17 October 2018). After removing the reads aligned to the ribosome RNAs, RF reads were mapped to the human genome sequence GRCh37 downloaded from Ensembl public database: http://www.ensembl.org accessed on 17 October 2018 using HISAT2 with default parameters [25,26]. We only used the uniquely aligned reads for further analysis.

Total mRNA sequencing reads were first aligned to the virus genome as done for the ribosome footprint reads. Then the unmapped reads were aligned to the GRCh37 reference using HISAT2 [25,26]. Similarly, as RF reads alignment, we performed the splice alignment for the paired-end mRNA-seq datasets with the default parameters. We only kept the uniquely aligned reads for the downstream analysis. The virus genome alignment quantification was done using featureCounts with the virus annotation (for both RF and mRNA sequencing in both ZIKV and DENV genome) [27]. The human genome alignment quantification for both RF and mRNA sequencing was done using featureCounts with the annotations of the protein-coding genes of GRCh37 as input. Only reads aligned to the exonic regions of the protein-coding genes were used for the downstream analysis using RiboDiff [28].

### 3.4. Footprint Profile Analysis Using Ribo-Diff

We used Ribo-diff to analyze the translation efficiency based on the ribosome footprinting and mRNA sequencing data [28]. Genes with at least 10 normalized read count as the sum of RF and RNA sequencing data were used as input, which resulted in 19,821 protein-coding mRNA. Genes with significantly changed translation efficiency were defined by the q-value cut-off equal to 0.05.

### 3.5. Real-Time PCR Assay

Total RNA was extracted using AllPrep DNA/RNA/Protein Mini Kit (Qiagen, Hilden, Germany, 80004). cDNAs were synthesized from 1 μg of total RNA using SuperScript III First-Strand (Invitrogen, Waltham, MA, USA, 18080-400) and were amplified using Taqman Universal Master Mix II, no UNG (Applied Biosystems, Waltham, MA, USA, 4427788). Analysis was performed by ΔΔCt. Applied Biosystems Taqman Gene Expression Assays: human SHMT2 Hs01059263_g1, PIM1 Hs01065498-m1, ATF3 Hs00231069-m1, and Beta-Actin 4332645. Relative mRNA expression was evaluated after normalization for Beta-Actin expression. Data show results from three independent experiments.

### 3.6. Motif Analysis

The longest transcript was selected to represent each corresponding gene. The 5′UTR sequences of the transcripts were collected for predicting motifs. Both the significant genes with increased or decreased TE and the corresponding background gene sets were used to predict motifs by DREME [29]. The occurrences of the significant motifs (E < 0.05 and *p* < 1 × 10^−8^ from DREME) were called using the FIMO [29] with default parameters for strand-specific prediction of all the 5′UTR sequences.

### 3.7. Statistical Analysis

All the results were analyzed with two-tailed t-tests unless specified. The significance of motif enrichments was from the DREME program based on the Fisher’s Exact Test. A hypergeometric test was performed to test for the significance of the enrichment of the gene overlap in GSEA pathway analysis.

## 4. Results

### 4.1. Transcriptional Changes Induced by ZIKV and DENV

We simultaneously sequenced total RNA and ribosome-protected RNA fragments from uninfected and virus-infected human neuronal progenitor cells (hNPCs) (Figure 1A, the complete dataset is submitted to GEO). Briefly, two independent clones of hNPCs were differentiated from hiPSC (n = 2) and infected with ZIKV (IbH isolate) and DENV-2 (strain 16681) (referred to as ZIKV and DENV from here on) with an MOI of 1 [23]. hNPCs were differentiated from hiPSC from different healthy donors (n = 2) and complete differentiation was characterized by immunostaining for Nestin and Sox9 (Appendix A). Consistent with prior observations, we found that ZIKV was more infective than DENV in hNPCs [30] and therefore we optimized the infection conditions to achieve equal infection rates as indicated by immunostaining for E-protein for both ZIKV and DENV (Appendix A). Quality control analysis of the RNA-seq data showed a good correlation between the uninfected and infected replicates (Appendix A). The read mapping analysis revealed around 5–17 million reads mapped to the human genome (hg19) and 1–4 million reads mapped to the ZIKV or DENV genomes in the respective samples (Appendix A and Appendix A). ZIKV infection resulted in upregulation of 445 mRNAs (q < 0.05) and downregulation of 335 mRNAs (q < 0.05) (Figure 1B, Appendix A), and DENV infection in upregulation of 156 mRNA (q < 0.05) and downregulation of 37 mRNAs (q < 0.05) in hNPCs (Figure 1C, Appendix A). A comparison of downregulated mRNA showed 26 mRNAs affected by both ZIKV and DENV, and 310 mRNAs or 11 mRNAs being exclusively downregulated by ZIKV and DENV, respectively (Figure 1D). A total of 112 mRNAs were upregulated by both ZIKV and DENV, while 333 mRNAs and 44 mRNAs were exclusively upregulated by ZIKV and DENV, respectively (Figure 1E). These data indicate overlapping and distinct effects of ZIKV and DENV infection.

### 4.2. Signature of ZIKV and DENV Transcriptional Repression

STRING analysis of genes whose expression was decreased in ZIKV infected cells (n = 335 revealed three major clusters (Figure 1F), with genes outside of the two main clusters making up cluster III and falling into two sub-clusters (Figure 1G) (PPI enrichment *p*-value < 1.0 × 10^−16^). Cluster I (n = 66) consisted mainly of histone genes localized to Chr. 6p22 [31] (Figure 1H) (PPI enrichment *p*-value < 1.0 × 10^−16^). The smaller Cluster II (n = 18) is composed of immune response genes (grouped as a systemic lupus signature), and Cluster III (n = 250) included many cell-cycle and mitosis-related genes (Figure 1I and Appendix A). Transcription factor binding site analysis showed a significant (*p*-value < 6.7 × 10^−14^ and q value < 5.6 × 10^−12^) enrichment of motifs related to NFY, TATA, and members of OCT and FOXO transcription factors (Figure 1J). The effects of DENV infection were quite distinct. The 37 genes down-regulated by DENV infection included PI3K-AKT-mTOR pathway genes such as SESN3, PI3KR1, and PI3KR3 (Figure 1K). KEGG analysis showed an enrichment (*p*-value < 1.3 × 10^−3^ and q value < 8.4 × 10^−3^) of proliferative pathways related to mTOR and the MYC/E2F transcription programs (Figure 1L). Hence, ZIKV and DENV-infected cells showed downregulation in histone gene expression and proliferation signatures consistent with impaired cell growth.

### 4.3. ZIKV and DENV Infection Activate Distinct Transcriptional Programs

ZIKV infection resulted in the upregulation of 445 mRNAs (q < 0.05) and DENV upregulated 156 mRNAs (q < 0.05) (Figure 2A,B). KEGG analysis showed that both viruses broadly activated expression programs related to aminoacyl tRNA synthetases (ARS genes) along with MAPK and p53 signals (Figure 2C–E). ZIKV infection further increased the expression of the constitutively active PIM1 kinase that stimulates translation and neurotrophin signaling, and these have been implicated in ZIKV replication [32,33] (Figure 2C,G,H and Appendix A, ZIKV targets in red). DENV infection-induced expression of enzymes related to the one-carbon metabolism (e.g., serine hydroxymethyltransferase-2 (SHMT2), methylenetetrahydrofolate dehydrogenase (MTHFD1L), which provide activated methyl groups in the form of S-adenosylmethionine (SAM) for nucleotide synthesis and post-translational modifications (Figure 2D,I, DENV targets in red). RNA expression of SHMT2, PIM1 and ATF3 has significantly upregulated upon ZIKV and DENV infection as observed by qRT PCR (Figure 2F). Furthermore, these effects on transcription correspond to different enrichment of transcription factor binding sites in deregulated genes enriched in the ZIKV compared to DENV infected cells. For example, ZIKV-induced transcriptional changes indicate a role for BACH1/2, whereas DENV infection appears to alter the CHOP/ATF3 transcription program (Figure 2I,J and Appendix A). Other transcription factor binding sites are equally represented and include AP1, ETS2, MAZ, SP1, and NFAT (Figure 2I,J). Together, the expression data reflect distinct cell stress responses triggered by each virus, and they also suggest strategies to augment RNA translation and replication through PIM1, ARS-mediated tRNA loading, and S-adenosylmethionine (SAM) production.

### 4.4. Measuring Translational Effects of Flavivirus Infection

We measured effects on host cell mRNA translation by ribosome profiling on ZIKV and DENV infected hNPCs (MOI = 1) at 72 h post-infection in duplicates. Briefly, we used our published method of RiboDiff to measure the translation efficiency of transcripts that are differentially regulated upon virus infection [28]. A summary of reading counts mapped to ribosomal RNAs, virus genome, and the human genome is provided in Appendix A. On average, 4.4 million RF reads mapped to the coding region of the human genome in uninfected cells, 4.9 million in ZIKV infected, and 4.8 million in DENV infected hNPC samples, corresponding to coverage across 19,821 protein-coding genes. Quality control analysis of replicates showed significant correlations among the replicates with a Pearson coefficient >0.97 (Appendix A). We used the RiboDiff statistical framework to analyze changes in mRNA translation (38).

### 4.5. ZIKV Infection Alters the Translation of Polyamine Metabolism Enzymes

We examined host cell mRNAs whose translation was altered upon viral infection. Applying a statistical cut-off at FDR <5% (FDR of <10%), we identified 19 (58) repressed mRNAs and 6 (22) translationally augmented mRNAs in ZIKV infected hNPCs (Figure 3A, Appendix A). Applying the same stringent criteria, we identified 7 (16) repressed mRNAs and 19 (33) upregulated mRNAs in DENV-infected hNPCs (Figure 3B, Appendix A). While relatively few mRNAs have translational changes disproportional to changes in their transcription, we noticed that both viruses equally affect specific RNAs such as ST8SIA1 (TE up), RPS3A (TE up), and SMOX (TE down). These shared translational effects may point to important biological effects. For example, RPS3A (ribosomal protein S3a) is a component of the 40S ribosome and is critical for viral protein production [34], and SMOX (Spermine Oxidase) oxidizes natural polyamines such as spermine [35]. Notably, ZIKV also upregulated the translation of OAZ2 (Ornithine Decarboxylase Antizyme 2), a key regulator of ornithine decarboxylase that catalyzes the rate-limiting step of the polyamine biosynthesis [35]. A STRING functional protein association network analysis for the top translationally repressed genes in ZIKV infected cells (DNM2, ATXN2L, HDGFRP2, SMOX, BAG3, and GBF1) further suggests effects on membrane and transport processes related to endocytosis, COPI vesicle coating, and receptor uptake (Appendix A). Translationally upregulated genes include ST8SIA1, ATP5E, RPS3A, HIST2H2AC, SPCS3, and PTCH2 (Figure 3A). Among these, RPS3A and SPCS3 are notable for their known roles in the translation of flavivirus proteins and the virion production [36,37]. Hence, an unbiased assessment of translational changes induced by ZIKV infection reveals translational control of polyamine metabolism that is required for the unique hypusine modification of the eIF5A translation factor and has been implicated as a target for antiviral therapies [38,39,40].

### 4.6. DENV Shares Key Translational Effects with ZIKV

Analysis of DENV-infected cells showed repression of the translation of SOCS3, SMIM15, LSM7, TEF, SMOX, KIAA0195, and NFE2L1 (NRF1) (Figure 3B). STRING functional protein association network analysis links these effects to JAK-STAT signaling, polyamine metabolism, and RNA and protein stability (Appendix A). On the other hand, DENV increased the translation of several ribosomal proteins, translation factors (EEF1A1, EIF3L), and other genes (ST8SIA1, SEC61G, TPT1) (Figure 3B,D). Similar to ZIKV, DENV downregulated the translation of SMOX, the key enzyme involved in polyamine catabolism (Figure 3B). Hence, DENV and ZIKV infection share effects on polyamine metabolism and DENV has additional pronounced effects on key translation initiation and elongation factors.

### 4.7. RNA Regulatory Motifs Enriched in Translationally Dysregulated mRNAs

To identify cis-regulatory RNA motifs in the 5′UTRs of mRNAs that are translationally affected by ZIKV and DENV, we compared the TE down and TE up groups for both ZIKV and DENV datasets. We included RNAs with annotated 5′UTRs and compared the groups to each other and to a background list of equally expressed and annotated mRNAs that showed no significant change in their translation compared to the uninfected control sample. For ZIKV, the groups were TE up (n = 83 at *p* < 0.05, q < 0.3), TE down (n = 228 at *p* < 0.05, q < 0.3), and background (n = 302); for DENV TE, up (n = 69 at *p* < 0.05, q < 0.3), TE down (n = 67 at *p* < 0.05, q < 0.3), and background (n = 208). Despite the relatively small size of groups, we identified four significant (*p* < 1.0 × 10^−5^) motifs in the TE up and TE down mRNA subsets for each virus. A binding site analysis shows that these sites correspond to known RNA binding protein sites. For example, the enriched RNA sequence in the TE up group of ZIKV infected cells corresponded to YBX1 and YBX2 binding sites (Figure 3E,F and Appendix A). We speculate that these RNA binding proteins contribute to some of the translational changes seen in infected cells, although this is pending further biochemical confirmation.

### 4.8. Analysis of Translation Efficiencies for the ZIKV and DENV Viral Genomes

We mapped 114,775 reads to the ZIKV genome and 277,897 reads to the DENV genome representing 11- and 12-fold coverage of ZIKV and DENV genomes, respectively (Appendix A and Appendix A). Compared to host mRNA translation, ZIKV and DENV RNAs were the second and third most highly translated mRNAs in infected hNPCs (Figure 4A,B). The ZIKV (ZIKV-IbH) and DENV (DV-2-16681) genomes are ~10 Kb in length and encode a polyprotein that is post-translationally cleaved by the host and viral proteases (NCBI Reference Sequence: NC_012532.1) [41,42]. This is expected to produce equimolar amounts of proteins; however, ribosome frameshifting can lead to preferential production of specific proteins as shown for West Nile Virus [43,44]. In both viruses, RNA expression and translating fraction (Ribo read counts) are correlated (Pearson r = 0.76; Spearman r = 0.68) (Figure 4C,D). Detailed analysis of RNA and ribosomal read coverage across the viral genomes shows the variation that may reflect low read counts, technical biases, and ribosome stalling at specific sites [45] (Figure 4E,F). The DENV 3′UTR (453 bases) is highly abundant and shows low ribosome coverage, whereas the DENV 5′UTR (96 bases) shows high ribosome coverage (Figure 4D,F). In the DENV 5′UTR, we detect potential non-AUG start codons in only +1 and +2 frames, suggesting one or two upstream open reading frames (uORF) that precede the 0 frame start codon of the capsid protein (Appendix A). The annotated ZIKV-IbH isolates have a 5′UTR (106 bases) (Appendix A). A detailed analysis of AUGs with ribosomal coverage and Ribo/RNA ratio in the capsid protein reveals three potential ORFs indicated by a ribosome peak with ORF1 starting at the position 36 (AUG codon), the second ORF2 initiating at AUG (position 51), and a third ORF3 at AUG (position 81) (Appendix A). Similar to DENV, we detect high levels of RNA and Ribosome reads in the ZIKV 3′UTR (428 bases) RNA (Figure 4E,F and Appendix A) that has previously been implicated in repressing viral replication [46].

## 5. Discussion

We provide a detailed and unbiased analysis of the transcriptional and translational dynamics of ZIKV and DENV infection in human neuronal progenitor cells. Previous studies in different cell types have reported many effects on the interferon response [7], MX2-related transcription program in B cells [8], and changes in the expression of extracellular matrix proteins [9], cell cycle, RNA processing, and cell metabolism [10,11]. Our analysis highlights cellular stress responses to viral infection, and on the other hand, we observe the activation of mechanisms that support the viral life cycle. Regarding stress responses, our data indicate that DENV preferentially triggers an unfolded protein response (UPR) program related to the ATF3/CHOP/DDIT3 transcription factors, whereas ZIKV favors a different, BACH1/2-NRF2 driven antioxidant program. Importantly, this ZIKV-induced redox program has previously been implicated in facilitating ZIKV replication [47,48,49,50]. We see other changes that also appear to enhance viral replication and translation. For example, increased expression of rate-limiting one-carbon metabolism enzymes such as SHMT2 and MTHFD1L provides activated methyl groups that are needed for nucleotide biosynthesis and viral replication; these mechanisms have been studied in cancer, and inhibitors are available [51,52,53,54]. We also notice an increase in tRNA loading enzymes -aminoacyl-tRNA synthetase (ARS) and expression of the constitutive active PIM1 kinase that stimulates protein synthesis in an mTOR independent manner [32]. Notably, both ARS and PIM1 have recently been implicated in flavivirus and ZIKV translation and replication [33,55]. Hence, the gene expression changes reflect both cellular responses and viral survival strategies and support potential cellular targets in metabolism and translation as novel antiviral strategies.

The re-programming of protein synthesis away from host mRNAs towards viral protein synthesis is a particularly stunning aspect of viral biology [12,56,57]. Other studies have explored the complex biochemical mechanisms [12,56,57]. Our study confirms preferential translation of viral RNAs, and we further provide a catalog of translational changes that include potential opportunities for antiviral attack. For example, ZIKV and DENV infected cells show downregulation of SMOX translation, which will decrease polyamine catabolism and thus increase polyamine availability for viral replication and translation [58]. A recently reported polyamine prodrug is thought to act in the exact opposite manner and increase SMOX expression, thereby depleting the required metabolites [59,60]. This pathway has been implicated as a broad spectrum anti-viral strategy beyond ZIKV and DENV and, intriguingly, both viruses target the translation of a key polyamine catabolic enzyme [38,59,60,61,62,63,64]. We detect other translational effects that have been implicated in viral biology. For example, the ribosomal protein RPS3A stands out among translationally activated host mRNAs, and RPS3A has been shown to interact directly with the DENV NS1 protein and augment the viral RNA translation [36]. Similarly, ZIKV infected cells increase translation of the Signal Peptidase Complex Subunit 3 (SPCS3) mRNA which has been identified as a genetic requirement for virion production for several flaviviruses [37]. Together, we provide a detailed accounting of the transcriptional and translational effects of DENV and ZIKV infection, however, further follow-up and experimental validation of these effects are much needed in hNPCs and other models of DENV and ZIKV infection. The data presented here is based on high throughput sequencing studies that are statistically robust and provide descriptive data indicating the potential gene expression programs and translational changes induced by virus infection. Further investigation based on our analysis would help to underscore the accuracy and relevance of biochemical studies and may inform the development of targeted antiviral therapies by inhibiting host factors relevant for viral replication.

## Figures and Tables

**Figure 1 viruses-14-01418-f001:**
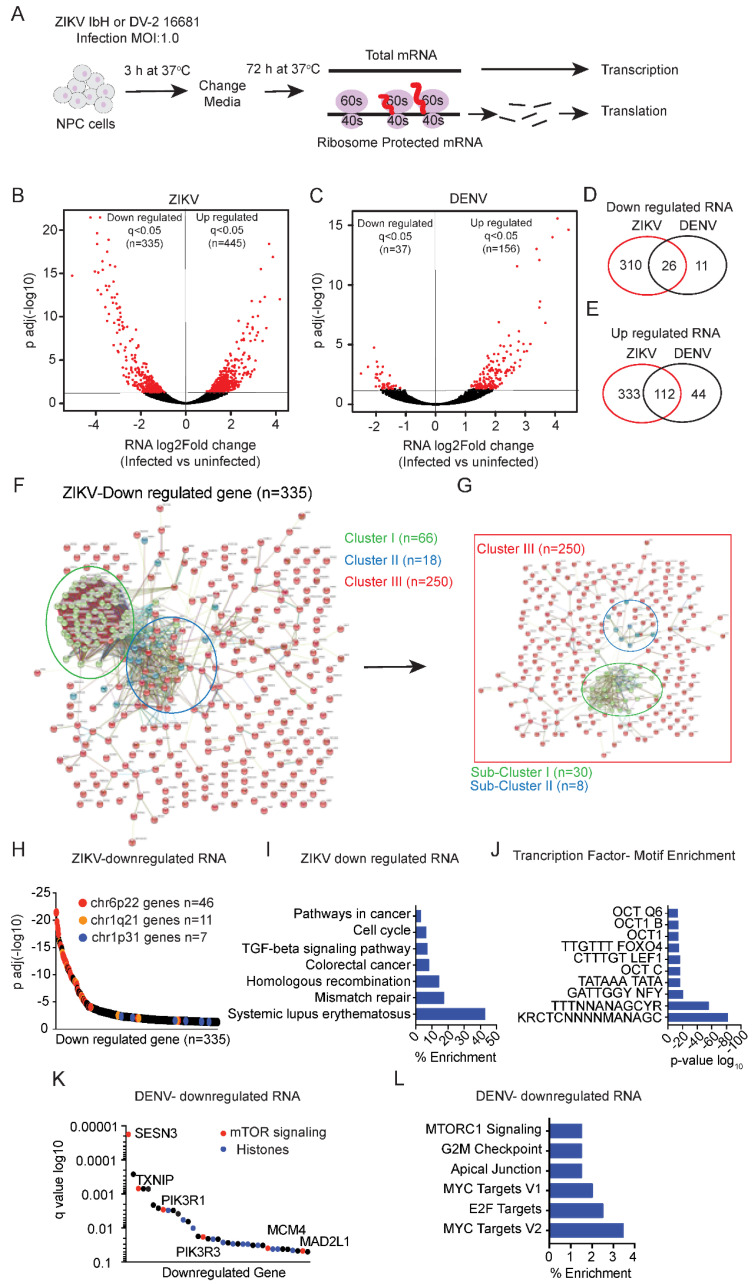
ZIKV and DENV induced transcriptional changes in hNPCs. (**A**) Schematic of the RNA-seq and ribosome footprinting on hNPCs cells infected with ZIKV or DENV (72 h). Comparison of ribosome-protected sequences vs. total mRNA isolates the translational efficiency for each mRNA (TE). (**B**,**C**), RNA-Seq analysis identifies changes in transcription in hNPCs cells infected with ZIKV (**B**) and DENV (**C**) compared to uninfected samples. Using the statistical cut-off of 5% FDR (False Discovery Rate), we identify differential mRNAs with significantly decreased or increased (shown in red; q < 0.05) or unchanged transcription (background, shown in black); two biological replicates. (**D**,**E**), Venn diagram showing the number of shared or exclusive RNAs that are significantly downregulated (**D**) or upregulated (**E**) (q < 0.05) in ZIKV and DENV infected hNPCs (**F**,**G**), STRING: Functional Protein Association Networks (version 11.0) analysis of downregulated RNAs (n = 335; q < 0.05) in ZIKV-infected hNPCs shows (**F**) three enriched clusters I, II, and III related to histone genes (green), SLE, viral carcinogenesis, alcoholism (blue), and cell cycle (red), respectively. (**G**) Cluster III showed sub-clusters related to the cell cycle mitotic (green) and resolution of sister chromatid pathways (blue). (**H**), Gene Set Enrichment Analysis (GSEA) for chromosomal positional sets showed significant enrichment of genes clustered on chr6p22, chr1q21, and chr1p31 among the downregulated RNA (n = 335; q < 0.05) in ZIKV infected hNPCs. (**I**), GSEA for KEGG pathway enrichment in the subset of mRNA downregulated upon ZIKV infection. (**J**), GSEA of transcription factor motif enrichment in the subset of mRNAs downregulated in ZIKV infected samples. (**K**,**L**), GSEA for KEGG pathways show significant enrichment of mTOR (red) (**K**), histone (blue) genes (**K**), and MYC/E2F targets (**L**) in downregulated mRNA (n = 37; q < 0.05) in DENV infected hNPCs.

**Figure 2 viruses-14-01418-f002:**
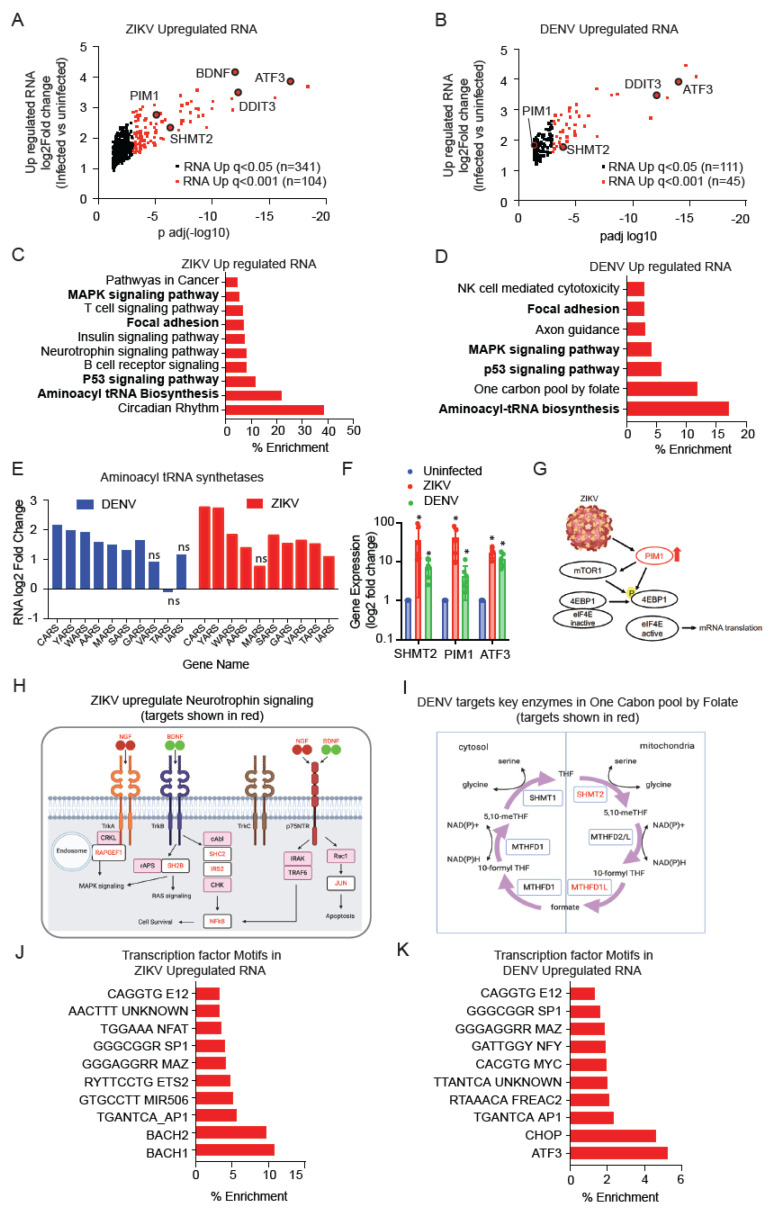
ZIKV and DENV infection-induced distinct transcription programs. (**A**,**B**), RNA-seq identified significantly up-regulated RNA at statistical cut off of q < 0.001 (red) and q < 0.05 (black) in ZIKV (**A**) and DENV (**B**) infected hNPCs. (**C**,**D**), GSEA for KEGG pathway enrichment in the subset of mRNA up-regulated upon ZIKV (**C**) and DENV (**D**) infection. Pathways enriched in both ZIKV and DENV infected hNPCs are indicated in bold. (**E**), RNA fold change of ZIKV and DENV induced aminoacyl tRNA synthetases (ARS) genes. (**F**), Gene expression analysis by qRT PCR for SHMT2, PIM1, and ATF3 in uninfected, ZIKV, and DENV infected hNPCs (biological replicates n = 3; experimental replicates n = 5–7; *p <* 0.05). Gene expression is normalized to beta-actin and fold change is plotted compared to uninfected samples. (**G**), ZIKV induced PIM1 expression that activates mTOR and mRNA translation. (**H**), ZIKV-induced transcription of genes involved in neurotrophin signaling, ZIKV targets are indicated in red. (**I**), DENV infection upregulates key enzymes involved in the one-carbon pool by the folate pathway, DENV targets are indicated in red. (**J**,**K**), GSEA transcription factor motif analysis identify common and distinct transcription factors involved in up-regulation of specific RNAs in ZIKV (n = 341; q < 0.05) (**J**) and DENV (n = 111; q < 0.05) (**K**) infected hNPCs.

**Figure 3 viruses-14-01418-f003:**
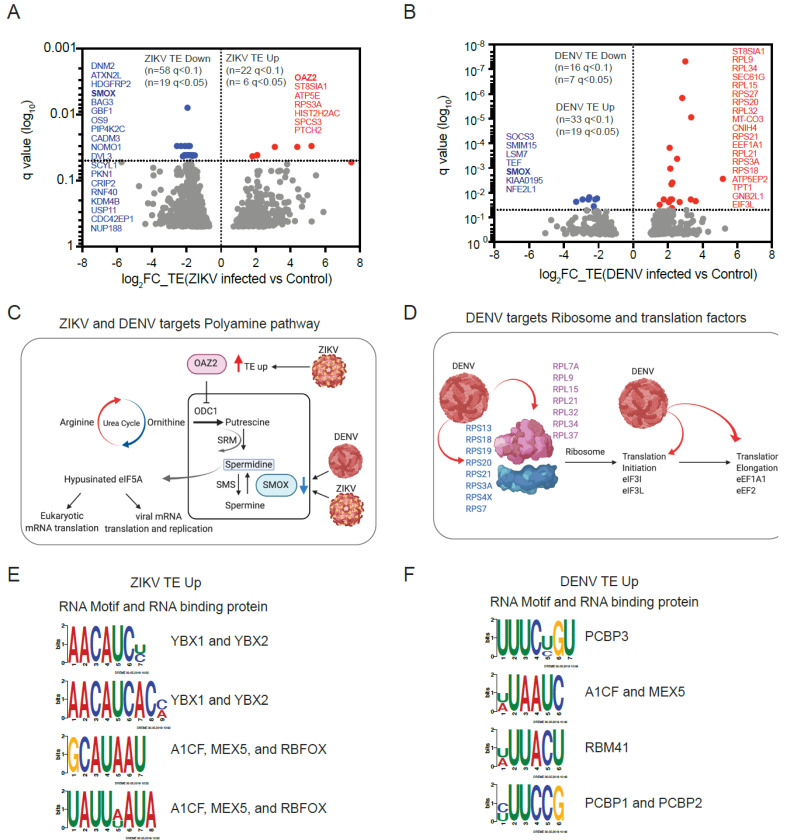
Translational changes induced by ZIKV and DENV infection. (**A**,**B**), Ribosome footprinting identifies a specific subset of mRNAs that significantly affected translation efficiency (TE) in ZIKV (**A**) and DENV (**B**) infected hNPCs. Using the indicated statistical cut-offs we identify mRNAs with decreased (TE down, red), increased (TE up, blue), and unchanged translation (background, grey); three biological replicates; the most significantly affected genes (q < 0.05) are indicated on each side. (**C**), ZIKV and DENV regulated the translation of key enzymes OAZ2 and SMOX in the polyamine pathway. (**D**), DENV upregulated multiple ribosomal proteins, translation initiation, and elongation factors. (**E**,**F**), Unbiased search for significantly enriched sequences (TE up versus background) identifies four motifs enriched in upregulated mRNAs in ZIKV (**E**) and DENV (**F**) infected hNPCs. RNA binding proteins associated with each motif are indicated.

**Figure 4 viruses-14-01418-f004:**
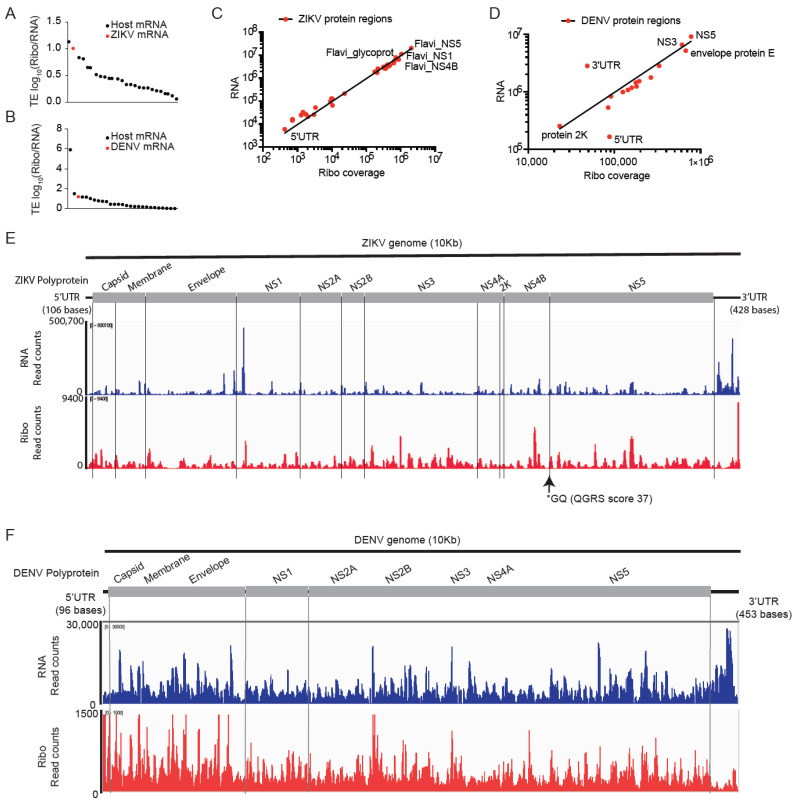
Analysis of translation efficiencies for the ZIKV and DENV viral genomes. (**A**,**B**), TE analysis revealed that ZIKV and DENV genomic RNAs (shown in red) are highly translated compared to host mRNAs. (**C**,**D**), RNA, and ribosome coverage on ZIKV (**C**) and DENV (**D**) showed a strong correlation except for the 5′ and 3′UTR regions of DENV (**D**). (**E**,**F**), RNA, and ribosome coverage across the ZIKV (**E**) and DENV (**F**) genome mapped to virus polyprotein. ZIKV and DENV showed relatively higher RNA reads at the 3′UTR and higher ribosomal coverage at 5′UTR suggesting differential RNA abundance and translation from UTR regions.

## Data Availability

Sequencing data generated in this study will be available in the NCBI, Gene Expression Omnibus database (the data accession number: GSE207347).

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
