# Peer review of "Transcriptional and Translational Dynamics of Zika and Dengue Virus Infection"

_viruses, 2022, doi:10.3390/v14071418_

Round 1

Reviewer 1 Report

I am satisfied with this revision and the responses from the authors.

Reviewer 2 Report

The authors have addressed the issue brought up in the second round of peer review and I find the manuscript to be improved.

This manuscript is a resubmission of an earlier submission. The following is a list of the peer review reports and author responses from that submission.

Round 1

Reviewer 1 Report

While I find the manuscript to be significantly improved, there is one issue that was brought up in the original set of critiques that remains to be effectively addressed:

  1. 4E:  The 3’ UTR of the complete ZIKV genome is 427 bases long, not 33 bases long as claimed by the authors (see NCBI Reference Sequence: NC_012532.1).  The use of an incomplete Genbank accession number is not an effective reason to warrant calling the 3’ UTR almost 400 bases shorter than it is.  The correct 3’UTR needs to be used in the analysis and I would predict that levels of the 3’ UTR will be high as was seen in DENV (as well as in other published reports of ZIKV sfRNA gene expression). 

Reviewer 2 Report

The authors have added new results and replied the questions. But more experiments results are still needed to support the results in the manuscript. Moreover, the accession number HQ234500.1 is partial CDS sequence of polyprotein gene of Zika virus isolate IbH_30656 (https://www.ncbi.nlm.nih.gov/nuccore/HQ234500.1). There is no 3'-UTR sequence according to the annotation. The authors should to provide the source of the Zika IbH genome sequence.